# *Perceive, Ground, Reason, and Act*: A BENCHMARK FOR GENERAL-PURPOSE VISUAL REPRESENTATION

## ABSTRACT

Current computer vision models, unlike the human visual system, cannot yet achieve general-purpose visual understanding. Existing efforts to create a general vision model are limited in the scope of assessed tasks and offer no overarching framework to perform them holistically. We present a new comprehensive benchmark, General-purpose Visual Understanding Evaluation (G-VUE), covering the full spectrum of visual cognitive abilities with four functional domains — *Perceive*, *Ground*, *Reason*, and *Act*. The four domains are embodied in 11 carefully curated tasks, from 3D reconstruction to visual reasoning and visual manipulation. Along with the benchmark, we provide a general encoder-decoder framework to allow for the evaluation of arbitrary visual representation on all 11 tasks. We evaluate various pre-trained visual representations with our framework and observe that (1) Transformer-based visual backbone generally outperforms CNN-based backbone on G-VUE, (2) visual representations from vision-language pre-training are superior to these with vision-only pre-training across visual tasks. With G-VUE, we provide a holistic evaluation standard to motivate research toward building general-purpose visual systems via obtaining a better, more general-purpose visual representation.

## 1 INTRODUCTION

The long-term goal of machine vision (Marr, 1982; Barrow and Tenenbaum, 1981; Ikeuchi and Hebert, 1996; Marr, 2010) is to build a general-purpose vision system that can perceive, understand, and react to visual inputs from unconstrained environments. The term *general-purpose* can be best understood through observing our own visual systems, which support various complex higher-order visual tasks, from edge detection and object recognition to visual navigation and manipulation, all while rooted in a common brain region that produces core visual representations.

On the other hand, even though the field of computer vision has been thriving with models that solve complex tasks increasingly well over the last decade (He et al., 2016; 2017; Ronneberger et al., 2015; Dosovitskiy et al., 2020), there is still a considerable gap between vision models and the aforementioned human visual systems. In particular, current vision models are mostly task-specific and often contain specialized components or architectures designed for a specific setting. They are also hindered by diverse input-output formats that vary from task to task. Recent works in self-supervised learning (Oord et al., 2018; He et al., 2020; Chen et al., 2020; Grill et al., 2020; He et al., 2021; Huang et al., 2021; Ma et al., 2022) explore more general visual representations by learning from enormously-sized visual domains. Advances in vision-language modeling (Lu et al., 2020; Hu and Singh, 2021; Cho et al., 2021; Gupta et al., 2021; Kamath et al., 2021; Wang et al., 2022) also seek to unify the vision tasks with a task-agnostic model architecture. However, these models are evaluated solely on traditional tasks like classification, detection, or vision-language understanding. It is still unclear whether current computer vision models are general-purpose like the human visual systems, as there is no overarching benchmark that covers all visual tasks holistically, from low-level perception to high-level reasoning and acting.

To facilitate the research toward the general-purpose vision, we present General-purpose Visual Understanding Evaluation (G-VUE) benchmark. We carefully curate 11 tasks from four functional domains that visual systems should support —*Perceive*, *Ground*, *Reason*, and *Act*— ordered by their cognitive complexity. These four domains cover the full spectrum of human visual tasks and

extend far beyond the standard set of visual tasks like detection or V-L grounding discussed in previous models or benchmarks. Specifically, *Perceive* tests a model's geometry understanding. *Ground* examines a model's acquisition of visual semantics. *Reason* probes a model's capacity for logical deduction and common sense reasoning. *Act* investigates a model's ability for planning and decision-making by learning visual policies. Fig. 1 illustrates our benchmark and Tab. 1 shows the task details. Principles and details about task and dataset selection are presented in Sec. 3 and Appendix B. G-VUE provides a test-bed for a general-purpose vision system and allows for fair comparisons between different visual representations over a full spectrum of visual tasks.

The diverse visual tasks in G-VUE require the model to navigate in a wide range of input and output formats, including image, text, bounding box, dense map, and policy. There is no existing model that can accomplish all these tasks. It is quite different from the Natural Language Processing (NLP) field, where most tasks (Wang et al., 2018; 2019) can be accomplished using the same sequence-to-sequence model based on RNN (Mikolov et al., 2010), LSTM (Hochreiter and Schmidhuber, 1997), or Transformer (Vaswani et al., 2017). Therefore, we carefully equip the visual tasks in G-VUE with a general encoder-decoder framework to mitigate the gaps among various visual tasks and accommodate arbitrary visual encoder, which makes it possible to evaluate any visual representation on these 11 tasks. Sec. 4 further describes the principles and details

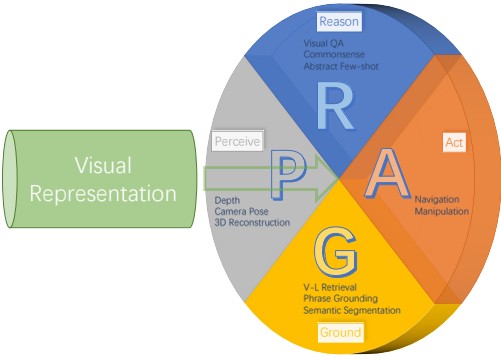

Figure 1: An overview of G-VUE. The key idea of G-VUE is to evaluate visual representation in a general-purpose standard, where P (*Perceive*), G (*Ground*), R (*Reason*), and A (*Act*) represent four functional domains for a general vision system.

of framework design. Notably, although such a framework facilitates handling tasks across various domains, it is not designed for massive multi-task training but for a flexible and fair evaluation of various existing visual representations on G-VUE.

To understand the challenges posed by G-VUE and the limitations of current models, we evaluate the most representative visual representations obtained with various architectures (*e.g.*, ResNet vs. ViT) and pre-training mechanisms (*e.g.*, supervised vs. self-supervised). The experimental results show that (1) Transformer-based architectures generally beat CNN-based architectures, and (2) visual representations pre-trained on massive image-text data (*e.g.*, CLIP) significantly outperform traditional ImageNet pre-training across visual tasks.

We make the following contributions: (1) we present a general-purpose vision benchmark G-VUE that consists of 11 visual tasks covering diverse visual skills; (2) we introduce an encoder-decoder framework for G-VUE that allows for the evaluation of arbitrary visual representations; (3) we examine the performance of state-of-the-art visual representations on G-VUE and present detailed analyses. Additionally, we construct an evaluation platform and a leaderboard for G-VUE that will be made publicly available upon acceptance.

## 2 RELATED WORK

### 2.1 VISUAL PRE-TRAINING

Large-scale visual pre-training has been viewed as a potential way to obtain a general visual representation that adapts well to any downstream tasks. There has been successful attempts at learning robust visual features from discriminative (Oord et al., 2018; He et al., 2020; Chen et al., 2020; Grill et al., 2020; Chen et al., 2021) and generative (He et al., 2021) self-supervised learning and vision-language alignment learning (Radford et al., 2021). These models use various visual backbones to learn from diverse visual domains, including images (Deng et al., 2009), egocentric videos (Grauman et al., 2022), and image-text pairs (Radford et al., 2021). Their differences in learning mechanism, visual backbone, and source learning domain affect their transferability to downstream tasks. Although these models have been extensively evaluated in existing tasks such as image classification and object detection, it remains unclear how they can be used to achieve a broader domain of visual tasks toward

general-purpose vision. One essential goal of G-VUE is to evaluate these models and provide tools for seeking more generalizable pre-trained models.

## 2.2 GENERAL-PURPOSE VISION

**Models**   The pursuit of general-purpose vision emerges from the early stage of computer vision. In the last 70s, Marr (2010) introduced a fixed layered representation and unified visual processing for general visual tasks. Despite suffering from its inflexibility to adapt to visual tasks, it provides a paradigm for general visual perception. Recently, progresses in deep learning drives the vision model towards unified architectures (Lu et al., 2020; Hu and Singh, 2021; Cho et al., 2021; Gupta et al., 2021; Kamath et al., 2021; Wang et al., 2022; Lu et al., 2022) that are agnostic to the input and output formats. Among them, VL-T5 (Cho et al., 2021) and GPV-1 (Gupta et al., 2021) unify the visual tasks through vision-language models, OFA (Wang et al., 2022) and Unified-IO (Lu et al., 2022) utilize tokenized representation to model various vision-language tasks jointly. Although these methods tackle a larger set of visual tasks, they are still limited to the conventional vision-language domain, lacking the general capabilities for low-level geometric understanding and high-level reasoning and acting.

**Benchmarks**   The general understanding benchmarks have become a standard in NLP (Wang et al., 2018; 2019) for testing pre-trained language models. However, few benchmarks are designed for general vision understanding, mainly due to the enormous complexity of diverse visual tasks. Among these benchmarks, Zhai et al. (2019) and Shao et al. (2021) focus on the transferability of visual representation, which perform domain adaptation by exposing pre-trained visual models to a small fraction of tuning data. GRIT (Gupta et al., 2022) emphasizes both generality and robustness of visual representation in visual perception and grounding tasks. Parisi et al. (2022) focuses on the evaluation of pre-trained visual representation on policy learning and explores how ingredients of pre-training affect the learning of control policy. While each of the previous benchmark only addresses a narrow aspect of human visual function, G-VUE provides holistic coverage of visual tasks spanning the functional domains of *Perceive*, *Ground*, *Reason*, and *Act*. With the equipped general encoder-decoder framework, G-VUE serves as a platform for evaluating any arbitrary visual encoder and representation in the context of general-purpose vision.

## 3   THE G-VUE BENCHMARK

Mirroring human cognitive abilities, our benchmark is comprised of tasks from four functional domains that a general vision system should support: *Perceive*, *Ground*, *Reason*, and *Act*. *Perceive* characterizes the basic ability of understanding geometry from raw visual input. *Ground* connects geometry with concepts and semantics. *Reason* operates on grounded objects and performs deductions. And finally, *Act* makes decisions and plans actions out of extensive reasoning. For each domain, we select visual tasks and datasets that are **representative, challenging, from the real-world domain, and available to the public**. The full list of the tasks is shown in Tab. 1. We provide an in-depth description in this section and Appendix B for task selection, dataset selection, and rationales.

### 3.1   PERCEIVE

Visual perception underlies human's subconscious mental approximation of geometric relationships in the physical world (Clements, 2004). In the domain of *Perceive*, we start with the evaluation of the sense of distance and space through **Depth estimation**. We move onto egocentric pose understanding with **Camera pose estimation**. Lastly, we examine the ability for detailed, shape-based geometric understanding with **Single-view 3D reconstruction**.

**Depth estimation.**   Monocular depth estimation is the task of estimating the egocentric distance to objects on a single 2D image. We adopt NYUv2 (Silberman et al., 2012), a common practice for evaluating monocular depth estimation. We make use of the updated version proposed in Lee et al. (2019) as the training set to provide a sufficient amount of data samples. The model's performance is evaluated by relative and absolute errors.

**Camera pose estimation.**   Learning-based absolute camera pose estimation is an instance-level task that requires the model to infer parameters about the point of view from a scene's geometry.

Table 1: The full list of tasks in G-VUE. We categorize our tasks according to their functional domains and provide details on the selected datasets, train/val/test splits, input/output modalities, and evaluation metrics.

| Task | Dataset | |Train| / |Val| / |Test| | Text | Output | Metrics |
|---|---|---|---|---|---|
| **Perceive** | | | | | |
| Depth Estimation | NYUv2 | 24k / - / 0.6k | N | Depth Map | d<1.25, AbsRel, RMSE |
| Camera Pose Estimation | CL & 7 Scenes | (3.8k, 26k) / - / (1.1k, 17k) | N | Camera Pose | Mean Trans. & Orient. Error |
| 3D Reconstruction | ShapeNetCore | 30k / - / 7.8k | N | Volumetric SDF | 3D IoU |
| **Ground** | | | | | |
| Image-Text Retrieval | Flickr30k | 29k / 1.0k / 1.0k | Y | Matching Score | Recall@1,5,10 |
| Phrase Grounding | RefCOCO | 42k / 3.8k / (2.0k, 1.8k) | Y | Bbox | Acc@0.5 |
| Semantic Segmentation | ADE20k | 20k / 2.0k / - | N | Segmentation Map | mIoU |
| **Reason** | | | | | |
| Question Answering | GQA | 943k / 132k / 12.5k | Y | Choice | Accuracy |
| Commonsense Reasoning | VCR | 213k / 26.5k / - | Y | Choice | Accuracy |
| Abstract Reasoning | Bongard-HOI | 23k / 17k / 14k | N | Binary Label | Accuracy |
| **Act** | | | | | |
| Navigation | R2R | 14k / (1.0k, 2.3k) / 4.2k | Y | Next Move | SPL |
| Manipulation | Ravens | 0.8k / 0.08k / 0.8k | Y | Pick & Place | Success Score |

Following the general setting in previous works (Kendall et al., 2015), we use both an outdoor urban localization dataset, Cambridge Landmarks (Kendall et al., 2015) and an indoor scenes dataset, 7 Scenes (Shotton et al., 2013) as our data source. We use the training and testing splits provided by Kendall et al. (2015), and report the mean errors of position (meters) and orientation (degrees) respectively over the scenes as our metric.

**Single-view 3D reconstruction.**    The task of single-view 3D reconstruction requires the model to predict a 3D voxel shape from a single 2D image. We evaluate on the synthetic images (Choy et al., 2016) of 3D objects from ShapeNetCore (Chang et al., 2015) and use the train/test split provided by Xu et al. (2019). We use 3D IoU in $64^3$ resolution as the metric to evaluate the performance.

## 3.2    GROUND

Visual grounding is defined as the process of connecting and aligning perceptual visual inputs with natural language. This alignment comes at different granularities, which form the basis of task progression in the *Ground* domain. We start with image-level content matching in **Image-text retrieval**, and proceed onto instance-level content matching in **Phrase grounding**. We further consider the composition of both granularities, *i.e.*, scene-level instance property grounding, in **Semantic segmentation**.

**Image-text retrieval.**    Image-text retrieval requires the model to match the image-text pairs given a sample from one modality. We select Flickr30k (Young et al., 2014) for this task. We follow the dataset split in Karpathy and Li (2015), with 29k/1k/1k images for training, validation, and test sets respectively. We use the recall of finding the correct caption in top-1 (R@1), top-5 (R@5), and top-10 (R@10) predictions as the metric for retrieval.

**Phrase grounding.**    In phrase grounding, a model is asked to predict a bounding box that crops out an object referred to by a natural language phrase. We select RefCOCO (Yu et al., 2016) as our source. It is a dataset built upon MSCOCO (Lin et al., 2014) with objects and bounding boxes labeled for each image. We evaluate on the benchmarking splits of person reference (testA) and non-person reference (testB) in RefCOCO, in addition to the validation set.

**Semantic segmentation.**    Semantic segmentation requires the model to ground semantic labels to pixels in each image and make dense predictions. For this task, we select the curated version of ADE20K (Zhou et al., 2017) and the corresponding train/test split from the MIT Scene Segmentation Benchmark (Zhou et al., 2017). It contains 150 categories of stuff (*e.g.*, sky) and discrete objects (*e.g.*, car) with their corresponding semantic masks on each image. We use the mean Intersection over Union (mIoU) between predicted and ground truth segmentation mask as the evaluation metric.

## 3.3 REASON

Visual reasoning characterizes the process of collecting grounded facts from images and making a complex deduction, *e.g.*, answering questions. In *Reason* tasks, we first visit **Visual question answering**, a seminal task that comprehensively examines whether a learner can jointly reason with images and text. We then advance to visual **Commonsense reasoning**, where a model also needs to acquire physical and societal commonsense through learning. Our final task examines a not often discussed niche in visual tasks: **Abstract and few-shot reasoning**.

**Visual question answering.** In a VQA problem, the model needs to generate the answer to a question (a single word or a compound word) after viewing an image. We select GQA (Hudson and Manning, 2019) for this task as it requires challenging relational and multi-step reasoning. By default, GQA provides image region features extracted with Faster-RCNN and ResNet-50. We instead use grid features output by the backbone directly without object detection. GQA also offers rich metrics for evaluation, but we only use the overall accuracy in our benchmark to ensure consistency with other tasks. Accuracy on the test-dev set is reported.

**Commonsense reasoning.** This task aims at answering multiple-choice questions out of scenarios that require commonsense knowledge. To crack these puzzles, a learner needs to understand human states, activities, social norms, and other commonsense knowledge. We select VCR (Zellers et al., 2019) dataset to evaluate commonsense reasoning with images. In this dataset, we follow the data preprocessing pipeline in Zellers et al. (2021), referring to entities by highlighting regions with pre-defined colors. For simplicity, we only incorporate the question answering (Q → A) subtask and report the overall accuracy on the validation set.

**Abstract and few-shot reasoning.** We employ Bongard-HOI (Jiang et al., 2022) as a proxy to evaluate the ability of abstract and few-shot reasoning. The task can be briefly described as follows: given a small set of positive and negative images that either depict or don't depict a human-object interaction (HOI) concept, the objective is to determine whether the additionally provided query image depicts that HOI concept. Bongard-HOI offers four val and test sets, respectively, which comprise seen/unseen objects and seen/unseen actions, allowing systematic evaluation of the model's generalization capability. We follow the evaluation protocol introduced in Jiang et al. (2022) and evaluate the overall accuracy on the four test sets.

## 3.4 ACT

*Act* is the most complex function vision systems could support. We consider two aspects of acting: navigating in an environment and manipulating objects. Both tasks require extensive reasoning about the environment and planning accordingly. Specifically, **Navigation** proposes the challenge of identifying one's location, understanding their possible movements, and directing to the target location from an egocentric perspective. **Manipulation**, on the other hand, focuses on learning from allocentric-view observations for planning with granular visual-motor skills.

**Navigation.** We adopt the Matterport3D Simulator (MatterSim) (Anderson et al., 2018a) and its associated Room to Room (R2R) dataset as the setting for vision-language navigation (VLN). This task requires the agent to follow natural language instructions and traverse a virtual house on a mesh formed by viewpoint locations, each with a panoramic view of the surroundings. Following previous works, we evaluate the performance on validation subsets and use SPL (Anderson et al., 2018b) in the unseen environment as the primary metric.

**Manipulation.** We follow CLIPort (Shridhar et al., 2022) and adopt their environment for the manipulation task. This task requires the agent to perform compositional actions (*e.g.*, pick and place) based on visual observations and language instructions. As in Shridhar et al. (2022), visual modules make affordance predictions to guide the subsequent action module on where and how to manipulate. We collect 8 subtasks with seen/unseen splits out of the 10 initial language-goal-conditioned subtasks proposed by Shridhar et al. (2022) and perform affordance learning on seen split of subtasks jointly, with 100 demonstrations for each task. We adopt the success scores used in Zeng et al. (2020) and report average scores on the unseen subsets of subtasks for evaluation.

## 4 FRAMEWORK

Due to the diverse nature of tasks in G-VUE, evaluating visual representations could be challenging. To this end, we propose an encoder-decoder framework for the visual tasks, making it possible to adapt arbitrary visual representations to all 11 tasks.

An overview of our framework can be found in Fig. 3. Firstly, the visual representation is viewed as the output of an image encoder, *e.g.*, ResNet-50. For tasks that also incorporate textual input, a text encoder is used to produce appropriate language embedding. Specifically, we employ RoBERTa (Liu et al., 2019) for language-involved tasks. Then the visual representation (and possibly the language embedding) will be sent to a task decoder. Since the tasks in G-VUE require various formats of output, we design different decoders for each class of output format. We briefly summarize the variants of our decoder as follows. More details are deferred to the Appendix C.

- **Label decoder.** This decoder is composed of several Transformer blocks and consumes tokenized visual representation (possibly concatenated with text embedding). We insert an extra `[CLS]` token for prediction. It works with most of the classification and regression tasks in G-VUE.
- **Dense decoder.** This decoder adopts the Segformer head (Xie et al., 2021), a light-weight module that maps tokenized visual representation to dense output. We use it in dense prediction tasks.
- **3D decoder.** This decoder resembles the pipeline in Mittal et al. (2022), where a Transformer-based decoder maps tokenized visual representation to a joint distribution over shapes and finally produces volumetric SDF via a pre-trained VQ-VAE. It is only invoked in the 3D reconstruction task.
- **Navigation decoder.** This decoder follows Hong et al. (2021), which maps visual representations and instruction embeddings to movement actions in a recurrent manner.
- **Manipulation decoder.** We employ the design in Shridhar et al. (2022) and map the visual representation and instruction embeddings to an affordance map, which then guides a model-based planner to draw the next manipulation action.

Albeit our efforts to make the encoder-decoder framework more generic and powerful, we do acknowledge that many task-specific decoders could generate better results given the same visual representation. However, we would like to highlight that our design principle is to balance performance and the amount of inductive bias, as the purpose of G-VUE is to provide a generic and accessible platform for evaluating general-purpose visual representation. Architectures with too many complex and task-specific designs are not consistent with the design principle and long-term goal of G-VUE. More discussions can be found in the Appendix.

## 5 EXPERIMENT

### 5.1 VISUAL REPRESENTATIONS FOR EVALUATION

We analyze three factors that may affect the quality of visual representation: architecture, pre-training mechanism, and source data. In our experiments, we adopt two architectures commonly used in large-scale pretraining: ResNet (He et al., 2016) and ViT (Dosovitskiy et al., 2020). On top of these backbones, we consider variations of pre-training mechanisms including the extent of supervision (*e.g.*, supervised *vs.* self-supervised) and learning objectives (*e.g.*, discriminative (He et al., 2020) *vs.* generative (He et al., 2021)). For source data, we consider ImageNet (Deng et al., 2009), large-scale image-caption pairs (Radford et al., 2021), and Ego4D videos (Grauman et al., 2022). To fully explore the effect of these factors, we select 7 representative visual representations as shown in Tab. 2.

During training, we keep these representations fixed and tune task-specific decoders for each task. Notably, we mainly chose grid features in our evaluated visual representations, as it is more generic, flexible for adaptation, and efficient compared with object-centric features (Jiang et al., 2020; Kim et al., 2021). Recent studies (Carion et al., 2020; Caron et al., 2021; Ma et al., 2022) further shows emerging object-centric features in grid feature learning, which reveals their potential in tasks (*e.g.* VQA) where object-centric features are commonly adopted.

Table 2: An overview of evaluated visual representations.

| Representation | Architecture | Pre-training mechanism | Data |
|---|---|---|---|
| RN-IN | ResNet-50 | Supervised classification | ImageNet |
| RN-MoCo | ResNet-50 | Self-supervised Contrastive Learning | ImageNet |
| RN-CLIP | ResNet-50 | Vision-language Contrastive Learning | WebImageText |
| RN-Ego | ResNet-50 | Vision-language Contrastive Learning | Ego4D |
| ViT-32-CLIP | ViT-B/32 | Vision-language Contrastive Learning | WebImageText |
| ViT-16-CLIP | ViT-B/16 | Vision-language Contrastive Learning | WebImageText |
| ViT-16-MAE | ViT-B/16 | Self-supervised Masked Image Modeling | ImageNet |

Table 3: Quantitative results of visual representations on G-VUE. For space considerations, we use the abbreviation of words for identifying tasks (*e.g.* "Cam. Pose." for camera pose estimation, "I-T Retr." for image-to-text retrieval, "Phr. Grnd." for phrase grounding, "Sem. Seg." for semantic segmentation, "Com. Res" for common sense reasoning, "Abs. Res." for abstract reasoning, "Nav." for navigation and "Manip." for manipulation). Summary scores are attached at bottom, which only consider test-dev subset for score on VQA task, test subset for score on abstract reasoning task, and unseen SPL for score on navigation task. We highlight the best results on each task both in bold and with underline.

| Task | | RN-IN | RN-MoCo | RN-CLIP | RN-Ego | ViT-32-CLIP | ViT-16-CLIP | ViT-16-MAE |
|---|---|---|---|---|---|---|---|---|
| **Perceive** | | | | | | | | |
| Depth Est. | test d<1.25 ↑ | 0.6394 | 0.6400 | 0.6523 | 0.4548 | 0.7363 | 0.7608 | **0.7699** |
| | test AbsRel ↓ | 0.2190 | 0.2184 | 0.2097 | 0.3409 | 0.1699 | 0.1606 | **0.1554** |
| | test RMSE ↓ | 0.7292 | 0.7284 | 0.6987 | 1.0302 | 0.5888 | 0.5535 | **0.5486** |
| Cam. Pose | Trans.(CL) ↓ | 2.909 | **1.972** | 2.446 | 3.045 | 2.160 | 2.196 | 2.177 |
| | Orient.(CL) ↓ | 7.298 | 5.427 | 6.814 | 7.000 | 5.974 | 6.020 | **4.911** |
| | Trans.(7S) ↓ | 0.281 | **0.241** | 0.279 | 0.266 | 0.290 | 0.263 | 0.271 |
| | Orient.(7S) ↓ | 10.299 | 8.797 | 9.502 | 9.058 | 9.682 | 9.752 | **6.699** |
| 3D Recon. | test mIoU ↑ | 39.70 | 41.98 | 42.00 | 37.66 | 40.63 | 39.43 | **43.95** |
| **Ground** | | | | | | | | |
| I-T Retr. | test R@1 ↑ | 10.4 | 7.1 | 18.5 | 1.5 | 15.3 | **23.0** | 5.8 |
| | test R@5 ↑ | 27.2 | 22.5 | 42.2 | 4.7 | 38.6 | **50.9** | 20.5 |
| | test R@10 ↑ | 38.3 | 33.8 | 55.2 | 8.4 | 51.2 | **64.2** | 30.9 |
| Phr. Grnd. | val Acc@0.5 ↑ | 48.57 | 54.40 | 63.45 | 36.5 | **67.99** | 64.55 | 65.18 |
| | testA Acc@0.5 ↑ | 54.48 | 59.14 | 70.63 | 41.92 | **75.70** | 70.73 | 67.59 |
| | testB Acc@0.5 ↑ | 43.87 | 50.88 | 55.86 | 31.38 | 61.16 | 57.24 | **62.10** |
| Sem. Seg. | val mIoU ↑ | 18.83 | 18.05 | 22.32 | 5.00 | 29.14 | **35.86** | 26.25 |
| **Reason** | | | | | | | | |
| VQA | val Acc. ↑ | 52.79 | 50.20 | 56.09 | 45.56 | **59.41** | 59.18 | 54.82 |
| | test-dev Acc. ↑ | 47.54 | 45.02 | 50.01 | 41.20 | **51.88** | 51.65 | 48.50 |
| Com. Res. | val Acc. ↑ | 52.61 | 50.98 | 57.30 | 52.79 | 61.94 | **64.11** | 60.76 |
| Abs. Res. | val Acc. ↑ | 59.85 | 62.38 | 64.54 | 55.72 | 63.24 | **65.65** | 62.12 |
| | test Acc. ↑ | 62.14 | 64.63 | 67.32 | 56.09 | 65.55 | **69.43** | 62.85 |
| **Act** | | | | | | | | |
| Nav. | Unseen Succ. ↑ | 48.62 | 48.49 | 49.00 | 43.89 | 49.21 | 47.13 | **49.55** |
| | Unseen SPL ↑ | 42.61 | 43.49 | 43.49 | 37.88 | **43.99** | 43.60 | 43.72 |
| Manip. | Unseen Score ↑ | 32.19 | 36.94 | 37.47 | 27.78 | 28.80 | **40.80** | 37.98 |
| **Summary Score** | | 40.23 | 41.24 | 44.96 | 32.86 | 45.44 | **48.61** | 44.62 |

## 5.2 RESULTS AND ANALYSES

We present quantitative experiment results in Tab. 3. In addition to metrics on each task, we provide a *summary score* (the bottom line of Tab. 3) as an overall measure across all tasks by converting all error metrics into percentages and taking an average. We summarize our major findings as follows.

**Architecture.** In general, ***ViT outperforms ResNet on most visual tasks.*** This confirms similar observations made by a few recent works (Liu et al., 2021; Wang et al., 2021; Shen et al., 2021; Ma

et al., 2022). Specifically, we find ViT-32-CLIP and ViT-16-CLIP significantly outperform RN-CLIP on dense prediction tasks, *i.e.*, depth estimation and semantic segmentation. This may indicate Transformer architecture is endowed with better compatibility on fine-grained discriminative tasks.

**Pre-training mechanism.**    We compare three pre-training mechanisms: supervised learning, self-supervised learning, and vision-language contrastive learning. Supervised learning on ImageNet was once the most common practice for producing general visual representations, but it is recently challenged by the simple yet effective self-supervised learning (He et al., 2020; Chen et al., 2020). We observe similar results on G-VUE. Representation from supervised pre-training (RN-IN) is only better than self-supervised one (RN-MoCo) in a handful of tasks that require semantics (*e.g.*, image-text retrieval) while significantly inferior to self-supervised one in tasks highly dependent on geometric perception (*e.g.*, camera pose estimation).

Vision-language contrastive learning is another competitive pre-training scheme equipped with semantics alignment. Due to the contribution of language, visual representation obtained in this way shows considerably stronger generalizability regardless of the lack of direct supervision (Radford et al., 2021). This is demonstrated by the observation that ***CLIP visual representation dominates most V-L tasks, and achieves the highest summary score.***

**Source training data.**    The experimental results also indicate that, on visual tasks that require semantics, especially language alignment, the gap between representations trained from supervised learning and self-supervised learning on ImageNet could be less significant than the gap with CLIP visual representations. We hypothesize that the cause lies in the source training data. It appears that ImageNet is already insufficient to support the learning of a general-purpose visual representation that can accommodate the broad spectrum of tasks in G-VUE. On the other hand, ***the success of CLIP visual representations in G-VUE supports the recent paradigm shift to multi-modal pre-training,*** which leverages massive source data (*e.g.*, WebImageText) and facilitates better representation with grounded concept learning. Such large-scale pre-training provides a promising path towards general-purpose visual representation and could potentially help to solve G-VUE. As a side note, we also notice that visual representations learned from egocentric video clips (RN-Ego) fail to achieve comparable performances on G-VUE. The drastic domain shift from the hand-centered Ego4D data to more general natural images in G-VUE might be the primary cause for such performance drop.

**Language representation.**    To make the evaluation of visual representations fair, we provide them with an identical language representation, *i.e.*, RoBERTa as language encoder. Nevertheless, using language representation pre-trained on the text-only corpus to collaborate with visual representations may cause sub-optimal performances, see Appendix D. This is similar to the shortcoming of vision-only pre-training. Check Appendix D for detailed ablation studies on the language representation.

In addition, we explore some other factors that could improve the performances of visual representations on G-VUE, including fine-tuning and resolution. We summarize these studies to Appendix D.

## 5.3 TASK CORRELATION AND GENERALIZATION

It is believed difficult for visual representations learned from specific tasks to adapt to other domains. However, as our results suggest, this is not necessary the case (*e.g.* MAE models perform surprisingly well on *Ground* and *Reason* tasks with reconstruction objectives). Such findings further motivates the question: are there shared requirements on representations over the seemingly divergent tasks? As there is no clear relationship between tasks, we empirically estimate task correlations with our experimental results. Specifically, we construct a task correlation matrix by treating the representation's performance on each task as a variable. As shown in Fig. 2, we observe that most tasks show high correlations. Compared with

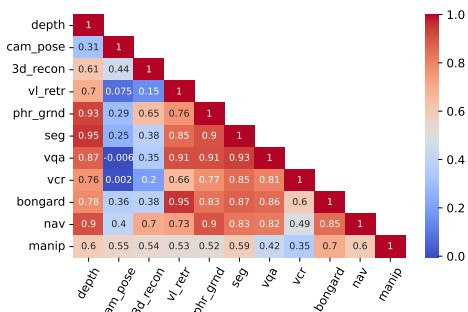

Figure 2: Correlations of 11 tasks in G-VUE.

other tasks, camera pose estimation and 3D reconstruction show fewer connections to other tasks. Nonetheless, camera pose estimation still correlates with manipulation since they both require 3D

Table 4: Evaluation of general-purpose vision-language models on G-VUE. For simplicity, we omit the tasks they cannot solve, *e.g.*, 3D reconstruction, navigation, and manipulation. *Note that performances on commonsense reasoning task are only for reference since models have no awareness of the referred objects from text description (which is adopted in Zellers et al. (2021) and G-VUE).

| Model | Depth / test | | | I-T retrieval | | | Phrase grounding / Acc@0.5 | | | VQA | Commonsense |
|-------|------|--------|------|------|------|------|-------|-------|-------|-------------|-------------|
|       | d<1.25 | AbsRel | RMSE | R@1 | R@5 | R@10 | val | testA | testB | test-dev Acc. | val Acc. |
| OFA-Huge | - | - | - | 26.1 | 46.3 | 54.9 | 92.04 | 94.03 | 88.44 | 59.92 | 28.19 |
| UIO-XL | 0.9439 | 0.0860 | 0.3505 | 33.0 | 53.9 | 62.8 | 86.17 | 86.28 | 85.30 | 36.91 | 32.09 |

geometric understanding. We also observe significant correlations in *Ground* and *Reason* domain. This result is intuitive since these tasks share similar objectives. We hope these findings could foster new insights in multi-task learning.

On the other hand, we find that visual representations may not be able to generalize across large modality gaps. RN-Ego is trained with first-person videos that have drastically different viewpoints compared to other data sources. As a result, we observe inferior performance of RN-Ego on G-VUE compared with other image-based representations. Additionally, we recognize that visual representations learned with strong biases are not suitable for generalizing across tasks, *e.g.*, representation fine-tuned on depth prediction shows significant performance drops when transferring to other tasks despite the high task correlation.

### 5.4 EVALUATION OF GENERAL VISION-LANGUAGE MODEL

Recently there emerges several unified vision-language model (Wang et al., 2022; Lu et al., 2022) that can perform multiple visual tasks by treating the task specification as language prompt and performing large-scale pre-training. To examine how general-purpose they are, we evaluate the OFA (Wang et al., 2022) and Unified-IO (Lu et al., 2022) on G-VUE. Specifically, we select their largest model variant and directly perform inference on tasks they can solve. Tab. 4 shows the results.

The evaluation results demonstrate the remarkable performance of recent unified models, particularly on tasks that they have been pre-trained. For example, Unified-IO achieves superior performances on depth estimation and phrase grounding, and both of OFA and Unified-IO reach the state-of-the-art on several VL tasks. We conclude that the large-scale multi-task pre-training significantly helps to elevate the performances on tasks that they have been pre-trained and enables certain level of generalization to various task prompt.

However, these models are still in lack of a general capability to deal with diverse tasks in G-VUE, *e.g.*, camera pose estimation, 3D reconstruction, abstract and few-shot learning, navigation, and manipulation. These visual tasks are substantial and form the core of human visual system, hence we advocate that the general vision model should expand to a broader domain, but not limited to vision-language tasks.

## 6 CONCLUSION

In this work, we present a novel general-purpose vision benchmark G-VUE, consisting of 11 meticulously chosen tasks. G-VUE covers the full spectrum of visual skills over four domains: *Perceive*, *Ground*, *Reason* and *Act*. We further introduce an encoder-decoder framework that supports the evaluation of arbitrary visual representations on all 11 tasks. With G-VUE, we evaluate the performance of state-of-the-art visual representations pre-trained with different learning paradigms and data sources. In particular, we find that Transformer-based architectures beat CNN-based models in most visual tasks and that pre-training on massive image-text pairs such as CLIP achieves significantly better results than ImageNet pre-training. Such findings shed light on the path to building a unified model for general-purpose vision.

We will also open-source an evaluation platform with a public leaderboard that can fairly and holistically evaluate different models on G-VUE. We hope this effort will promote the study of general-purpose representation learning and encourage the computer vision community to pursue universal, easily adaptable, and general-purpose visual models.

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

# A OVERVIEW

To present an intuitive visualization of our benchmark, we gather all the 11 tasks, as well as their inputs and outputs, and exhibit how the pipeline works in a holistic layout. See Fig. 3.

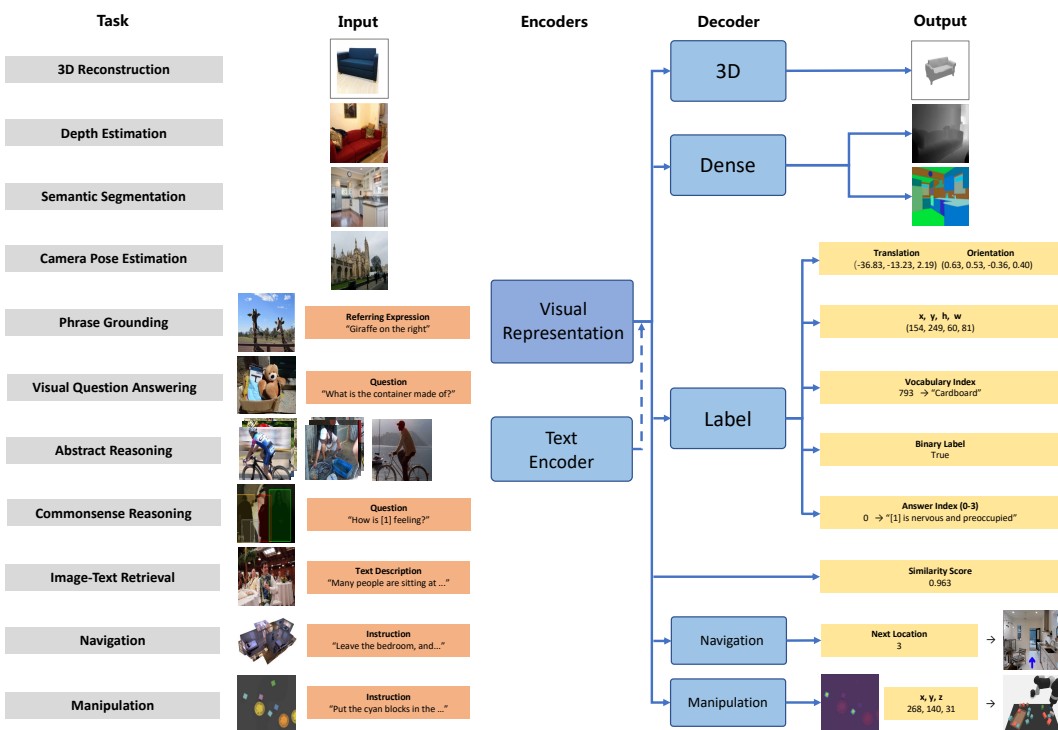

Figure 3: An illustration of our encoder-decoder framework. The visual representation is viewed as the output of an image encoder, e.g. ResNet-50. For tasks that also incorporate textual input, a text encoder is used to produce appropriate embeddings. The visual representation (and possibly the text embedding) will be sent to task decoders to accomplish full spectrum visual tasks.

# B DESIGN PRINCIPLE AND RATIONALITY

To present our detailed task and dataset selection process, we separate four domains and organize each domain by first delivering the motivations behind task selections and then explaining dataset choices.

## B.1 PERCEIVE

**Task.** Depth perception is the foundation of transition from 2D image to 3D space, which makes it essential for geometric perception. As such, this constitutes the first task in the *Perceive* domain. Specifically, we select monocular depth estimation, as it is a common practice. Apart from depth, there are many other tasks reflecting the capability of perception, most of which are covered in Zamir et al. (2018). Surface normal estimation is also a critical factor about scene geometry, but it has a large overlap with depth, *e.g.*, both are making a dense prediction. Line segment recognition is another one, but is ruled out for being too low-level. We would like to augment this domain with a task from a different perspective that can better incorporate scene-level properties, and thus camera pose estimation becomes our choice. Inferring the camera pose from a single image is crucial for egocentric downstream applications, *e.g.*, robotics.

With depth and camera pose, we are able to recover the scene layout from a macroscopic perspective. Next, we examine detailed perception on objects. One task candidate is 6D pose estimation, which requires the model to estimate both location and orientation of objects. However, it ignores some key

components in perception, *e.g*., the appearance and shape of objects. The ultimate goal of geometric understanding is to reconstruct objects, and this can further support 3D holistic scene reconstruction. Therefore, we choose single-view 3D reconstruction as a comprehensive test on *Perceive* domain.

**Dataset.** We choose NYUv2 dataset for evaluation on depth estimation task. Compared with KITTI (Geiger et al., 2012), another dataset often used for autonomous driving, NYUv2 covers more curated indoor scenes, which is consistent with tasks in *Act* domain. We choose the 7Scenes dataset and the Cambridge Landmarks dataset for evaluation on camera pose estimation. The former contains seven small-scale indoor scenes with only a few squared meters and the latter includes six large-scale outdoor scenes whose space ranges from 35x25m (Shop Façade) to 500x100m (Street). Regarding this choice, we consider two aspects: (1) they are widely adopted by previous works, (2) they encompass typical scenarios and challenges associated with estimating camera pose, *e.g*., scales, indoor scenes *vs*. outdoor scenes, and varying viewpoints. We select ShapeNetCore dataset for evaluation on 3D reconstruction task, because it is the most popular choice for single-view 3D reconstruction, and excellent materials and tools provided by the community can be utilized, including rendered images with diverse view directions, voxel grids of varying resolutions *etc*.

## B.2 GROUND

**Task.** In general, we develop this domain in an incremental manner. The most accessible multi-modal data are image-text pairs, and the identification of these pairs reflects the basic capability of grounding. Naturally, we select the image-text retrieval task to evaluate this capability.

The capability to match image-text pairs cannot guarantee correct spatial grounding. A complete vision model needs to tell not only what, but also where. Therefore we must consider the evaluation on correctly located objects. The reason for selecting phrase grounding instead of a conventional detection task is that we care more about precise recognition aligning with language. And since this task is similar to text-to-image manner, we implement the previous image-text retrieval task in image-to-text setting. To this end, we have instance-level grounding, expanding to object-phrase pairs.

But rectangle boundary prior in phrase grounding task limits the flexibility of grounding. Hence, We seek a finer-grained grounding task, which can be achieved by tagging each pixel. We select semantic segmentation rather than referring expression segmentation, since the ability to precisely ground a single object has been included in phrase grounding task and here we hope to evaluate holistic scene parsing. Such a skill is complementary to object-centered grounding. With these three tasks, we cover a rich diversity from several aspects, *e.g*., granularity, aligning with language *vs*. tagging, object-centered *vs*. scene-centered. Image captioning and human-object interaction (HOI) are also in our consideration, but the former has overlap with image-text retrieval and the latter is incorporated in Bongard-HOI.

**Dataset.** For I-T retrieval task, We choose Flickr30k, rather than MSCOCO (Lin et al., 2014) for the purpose of reducing data overlap with phrase grounding, which adopts RefCOCO dataset. RefCOCO is particularly proposed for referring expression comprehension task (also referred to as phrase grounding), incorporating cases where the model should be able to discriminate between similar instances according to subtle hints in language descriptions. RefCOCO+ (Yu et al., 2016) and RefCOCOg (Yu et al., 2016) are very similar to RefCOCO, so picking one of them is enough. We choose ADE20K for semantic segmentation because it is more challenging compared with other counterparts e.g. PASCAL (Everingham et al., 2010), Cityscapes (Cordts et al., 2016). And ADE20K also includes rich contents, *e.g*., stuff annotations like wall, sky *etc*.

## B.3 REASON

**Task.** For this domain, we consider vision tasks that require reasoning. As the evaluation of reasoning capability commonly comes in the form of question answering, we select visual question answering as the basis for task selection in this domain. Specifically, we consider question answering that requires different capabilities, starting from the simplest object grounding and relation inference, to commonsense reasoning, and finally to analogical reasoning between sets of images. We select this task based on their difficulties, where relational inference requires grounding capabilities within

the image, common sense reasoning requires common knowledge outside of the image, and finally, analogical reasoning requires finding similarities between different sets of images.

**Dataset.** To address the aforementioned critical dimensions of visual reasoning in the real-image domain, we select GQA, VCR, and Bongard-HOI as our benchmarking datasets. Specifically, we prefer the GQA dataset over the other commonly adopted dataset VQAv2 as it incorporates object grounding and more complex compositional reasoning over relationships of objects. VCR contains question answering problems on common sense knowledge. Bongard-HOI dataset is proposed for solving the Bongard-problem, where one needs to infer the common concept from a set of images and next rule out the odd one from the other set that does not follow the concept. Therefore, we adopt it as our benchmark for concept abstraction and analogical reasoning.

### B.4   ACT

**Task.** For this last domain, we consider visual tasks in an embodied setting. Navigation asks an agent to understand its positioning in an environment using visual information and planning movements to desirably move to a different place. Manipulation, on the other hand, requires the agent to achieve the desired configuration of the environment through planning the movement of environmental elements. We assume these two tasks can cover most embodied visual tasks.

**Dataset.** For navigation, we adopt the Matterport3D dataset and simulator to create virtual environment to train and evaluate virtual agents. We choose this dataset because it covers a real-world image domain with much more visual complexity, as well as a diversity of clean indoor environments, and is also frequently used for vision-language navigation compared to other navigation datasets like Habitat (Savva et al., 2019). For manipulation task, we follow CLIPort, leveraging the advantages of integrated methodology. The Ravens benchmark (Zeng et al., 2020) is proposed with rich ingredients to facilitate systematic evaluation. In addition, another important reason for our choices is that selected datasets and environments of the two tasks both provide seen and unseen splits for generalization testing.

## C   IMPLEMENTATION DETAILS

**Visual backbones.** We use Timm (Wightman, 2019) to build most of the visual backbones. Feature extractors are manually implemented if not supported. To obtain CNN-like feature pyramids from ViT backbones, we simply apply bilinear interpolation on intermediate layers as it shows little difference compared with transposed convolution. All ResNets are ResNet-50, and all ViTs are ViT-Base.

**Decoders.** We design three types of decoders for tasks other than *Act*, and their architectures are visualized in Fig. 4. Tasks in *Act*, *i.e.* navigation and manipulation, are performed by integrated agents in simulation environments, whose structures can be found in Hong et al. (2021) and Shridhar et al. (2022). The details of the decoders are as follows:

- **Label Decoder**. We implement the label decoder as a stack of Transformer blocks. The decoder flattens the last feature map from the encoder and then regards the feature as a sequence of tokens, to which fixed 2D positional embedding is applied. We concatenate the text token sequence with the visual feature sequence if needed. We insert an extra label token as the proxy to regress desired outputs. We pass this token through an MLP after Transformer layers for the final predictions. Note that we do not adopt this decoder in V-L retrieval task, where we instead model cosine similarity after processing embeddings of the two modalities through a simple MLP.
- **Dense Decoder**. We adopt the lightweight Segformer head (Xie et al., 2021) as our dense decoder to handle dense prediction tasks. It performs efficient spatial and channel-wise feature aggregation on the visual feature pyramid representation.
- **3D Decoder**. Implemented for the single-view 3D reconstruction task, the 3D decoder consists of three sub-decoders. The first up-convolution decoder maps 2D visual representation into a conditional distribution over the latent variables of 3D shapes. We then combine the predicted conditional distribution with autoregressive priors produced by a pre-trained Transformer-based decoder (Mittal et al., 2022) to obtain a joint distribution over 3D shapes. We finally obtain the volumetric SDF by forwarding the joint distribution to the third decoder, the decoder of a pre-trained Patch-wise Encoding VQ-VAE (Mittal et al., 2022). We freeze the weights except for the first sub-decoder.

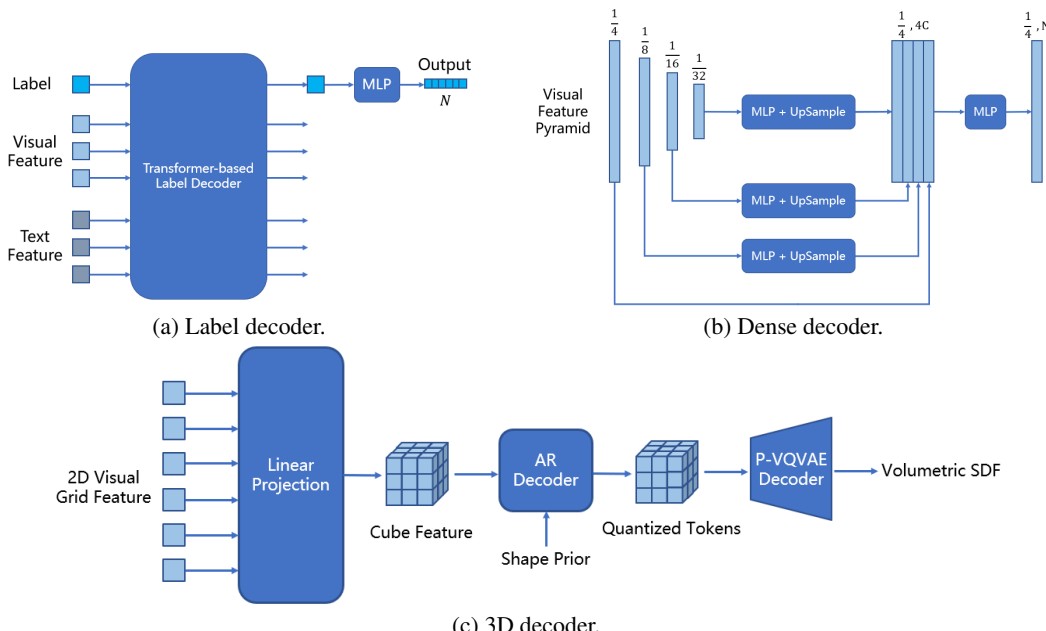

(a) Label decoder.

(b) Dense decoder.

(c) 3D decoder.

Figure 4: An illustration of the decoders. Let $N$ denote the output dimension, *e.g.* 4 for bounding box. (a) The output of Label Decoder is a single vector of length $N$. (b) The output of the Dense Decoder is a dense grid with each element being an $N$-length vector. This dense grid is then upsampled back to the original image dimension through a bilinear interpolation. (c) The output of 3D Decoder is a volumetric SDF. The Autoregressive(AR) decoder is a fixed module with pre-trained weights from Mittal et al. (2022).

- **Navigation**. Following Hong et al. (2021), we use visual backbones to pre-compute visual representations (using the last feature map) at every possible viewpoint in a given room setting to accelerate RL training. We then adopt Recurrent VLN Bert as the task decoder, specifically the PREVALENT-like (Hao et al., 2020) variant due to its superior performance over the original OSCAR-like (Li et al., 2020) model. And as in that paper, we initialize the decoder with pre-trained PREVALENT weights and use a combination of RL loss and imitation learning (IL) loss as the training objective.
- **Manipulation**. This decoder is inherited from CLIPort (Shridhar et al., 2022). Similar to the dense decoder, this decoder depends on the feature pyramid from visual representation and performs up-sampling at each level. Differently, language instructions are required to finish manipulation task. To process language input, the decoder tiles text features and aggregates them with visual features. The output of the decoder is a dense affordance map to guide the agent picking and placing. It is important to bring up that we remove the extra spatial stream to prevent the intervention of other visual representations, while the original CLIPort (Shridhar et al., 2022) includes that stream to attain improved performance.

Table 5: Settings for *Perceive*, *Ground*, and *Reason* tasks. *Act* tasks are omitted here since they belong to separate training schemes. Abbreviations: "Cam. Pose." for camera pose estimation, "I-T Retr." for image-to-text retrieval, "Phr. Grnd." for phrase grounding, "Sem. Seg." for semantic segmentation, "Com. Res" for common sense reasoning, "Abs. Res." for abstract reasoning.

| Setting | Depth Est. | Cam. Pose. | 3D Recon. | I-T Retr. | Phr. Grnd. | Sem. Seg. | VQA | Com. Res | Abs. Res. |
|---|---|---|---|---|---|---|---|---|---|
| Decoder | Dense | Label | 3D | MLP | Label | Dense | Label | Label | Label |
| Out Dim | 1 | 7 | 512 | - | 4 | 151 | 1843 | 4 | 2 |
| Batch Size | 128 | 32 | 32 | 256 | 128 | 128 | 128 | 128 | 32 |
| Epoch | 50 | 100 | 100 | 100 | 50 | 50 | 30 | 20 | 30 |

**Training.** Most of the training settings are listed in Tab. 5. We use AdamW (Loshchilov and Hutter, 2017) optimizer, with $\beta_1 = 0.9, \beta_2 = 0.99$ and weight decay at $10^{-2}$. During training, we warm up

the learning rate linearly in the first 2% steps to reach $1 \times 10^{-4}$ and adopt a linear decay schedule gradually decreasing the learning rate to zero during the rest of steps. The gradient norm is clipped to 1. Navigation and manipulation tasks are trained in simulation environments and stopped when convergence is observed. For simplicity, we resize input images to 224 to align with the resolution during pre-training phase of visual representations, which is necessary since these representations stay fixed during evaluation phase. All experiments are affordable on a single A100 GPU.

**Summary score.** To summarize the overall performance of a visual representation on various tasks and conclude how general-purpose it is, we propose a comprehensive summary score. However, performances on different tasks are in divergent metrics. In general, there are two types of metrics in our benchmark: percentage and error. We create a normalized mapping to convert all error metrics to percentage metrics: given a metric in error form $E$, the corresponding metric in percentage form $P$ is formulated as $P(E) = e^{-1.386E}$, where the constant $-1.386$ is determined by two conditions: $P(0) = 1$ and $P(0.5) = 0.5$. With this normalization, we can calculate a score for each task by averaging all aligned metrics within the task, and then calculate the final summary score for a visual representation by averaging scores on all 11 tasks. To make domains more balanced, we augment each task in *Act* with weight at 1.5.

## D    ABLATION

### D.1    FINETUNING

In the experiment section, we mainly report results of fixed visual representations, since the evaluation of general-purpose requires a shared visual representation for various tasks. To further study the effects of finetuning the visual backbones, we provide experiments on two visual representations learned in self-supervised fashion: RN-MoCo and ViT-16-MAE. During finetuning, we set the base learning rate of visual backbones to $10^{-5}$ and keep the same warm-up and linear decay schedule. As shown in Tab. 6, dense prediction tasks, *e.g.* depth estimation, benefit most from finetuning visual backbones. Other tasks in general can benefit marginally from finetuning. While a few cases turn out to be harmful finetuning. We hypothesize that learning decoder with a dynamic encoder is hard and prone to overfitting, especially on relatively small amount of training data.

### D.2    RESOLUTION

Another factor that may cause inferior performance is the resolution of the input image. We demonstrate this by showing performance gain from scaling up input resolution, see Tab. 7. The results reveal the headroom of performance when scaling up resolution.

### D.3    LANGUAGE ENCODER

**Image-text retrieval.** From the experimental results in Tab. 3, we find large variance on image-text retrieval task. As mentioned before, we hypothesize the reason for unsatisfactory performances on this task lies in the language encoder. With limited cross-modal processing in the implementation, the cross-modal matching highly depends on the quality of pre-trained representation. Since the default language encoder RoBERTa is pre-trained on text-only domain, the representation it produces may not align with vision branch well. We substitute it with the CLIP language encoder (Radford et al., 2021) pre-trained in vision-language contrastive learning manner. As we can see in Tab. 8, switching to the CLIP language encoder leads to significant improvements. This demonstrates the effectiveness of coupled representations, and also indicates the performance drop due to mismatching representations, though GPT2-CLIP and RoBERTa are both commonly recognized language encoders.

Actually, the brilliant performance of current state-of-the-art models on image-text retrieval task rely heavily on coupled representations obtained by large-scale vision-language pre-training. Besides, fixed language encoder initialing from text-only pre-training proves sub-optimal results on V-L matching task, as reported in Kim et al. (2021). To pursue general-purpose visual representation, how to overcome the gap between diverse disentangled representations of different modalities and provide fair yet efficient evaluation methods is still a challenging problem that remains unsolved.

Table 6: Influence of finetuning visual backbones. For space considerations, we use the abbreviation of words for identifying tasks (*e.g.* "Cam. Pose." for camera pose estimation, "I-T Retr." for image-to-text retrieval, "Phr. Grnd." for phrase grounding, "Sem. Seg." for semantic segmentation, "Com. Res" for common sense reasoning, "Abs. Res." for abstract reasoning). Finetuning on Act tasks is omitted here. We also remark the gains from finetuning in a separate column. Green indicates positive gain, while red indicates drop.

| Task | | RN-MoCo | | | ViT-16-MAE | | |
|---|---|---|---|---|---|---|---|
| | | Fixed | Finetune | Gain | Fixed | Finetune | Gain |
| **Perceive** | | | | | | | |
| Depth Est. | test d<1.25 ↑ | 0.6400 | 0.7302 | +0.0902 | 0.7699 | 0.8605 | +0.0906 |
| | test AbsRel ↓ | 0.2184 | 0.1725 | -0.0459 | 0.1554 | 0.1203 | -0.0351 |
| | test RMSE ↓ | 0.7284 | 0.6042 | -0.1242 | 0.5486 | 0.4444 | -0.1042 |
| Cam. Pose | Trans.(CL) ↓ | 1.972 | 1.943 | -0.029 | 2.177 | 1.926 | -0.251 |
| | Orient.(CL) ↓ | 5.427 | 5.671 | +0.244 | 4.911 | 3.837 | -1.074 |
| | Trans.(7S) ↓ | 0.241 | 0.255 | +0.014 | 0.271 | 0.250 | -0.021 |
| | Orient.(7S) ↓ | 8.797 | 8.452 | -0.345 | 6.699 | 5.823 | -0.876 |
| 3D Recon. | test mIoU ↑ | 41.98 | 42.87 | +0.89 | 43.95 | 48.25 | +4.30 |
| **Ground** | | | | | | | |
| I-T Retr. | test R@1 ↑ | 7.1 | 5.9 | -1.2 | 5.8 | 6.8 | +1.0 |
| | test R@5 ↑ | 22.5 | 17.3 | -5.2 | 20.5 | 24.1 | +3.6 |
| | test R@10 ↑ | 33.8 | 26.6 | -7.2 | 30.9 | 35.9 | +5.0 |
| Phr. Grnd. | val Acc@0.5 ↑ | 54.40 | 56.73 | +2.33 | 65.18 | 64.55 | -0.63 |
| | testA Acc@0.5 ↑ | 59.14 | 61.37 | +2.23 | 67.59 | 66.13 | -1.46 |
| | testB Acc@0.5 ↑ | 50.88 | 52.10 | +1.22 | 62.10 | 62.38 | +0.28 |
| Sem. Seg. | val mIoU ↑ | 18.05 | 27.40 | +9.35 | 26.25 | 35.21 | +8.96 |
| **Reason** | | | | | | | |
| VQA | val Acc. ↑ | 50.20 | 51.49 | +1.29 | 54.82 | 54.49 | -0.33 |
| | test Acc. ↑ | 45.02 | 45.44 | +0.42 | 48.50 | 47.89 | -0.61 |
| Com. Res. | val Acc. ↑ | 50.98 | 55.22 | +4.24 | 60.76 | 55.68 | -5.08 |
| Abs. Res. | val Acc. ↑ | 62.38 | 62.58 | +0.20 | 62.12 | 64.38 | +2.26 |
| | test Acc. ↑ | 64.63 | 62.36 | -2.27 | 62.85 | 65.14 | +2.29 |

Table 7: Performance progressively gains from fine-tuning and higher resolution. Take RN-IN on phrase grounding task for example. The results are reported in Acc@0.5 metrics.

| Setting | val | testA | testB |
|---|---|---|---|
| Fixed on 224 | 48.57 | 54.48 | 43.87 |
| Tuned on 224 | 58.65 | 64.15 | 53.26 |
| Tuned on 512 | 65.55 | 71.95 | 59.83 |

Table 8: Performance of different vision-language encoder pairs on Flickr30k test set.

| Encoder Pair (V+L) | Recall@1 | Recall@5 | Recall@10 |
|---|---|---|---|
| ViT-16-CLIP + RoBERTa | 23.0 | 50.9 | 64.2 |
| ViT-16-CLIP + GPT2-CLIP | 45.5 | 74.2 | 83.0 |

**Navigation.** Similar to Image-text retrieval, we also conduct ablation of language encoder on navigation task. It is customary in vision-language navigation task to use a pre-trained language encoder that produces visually-aligned representations to better aid navigation. However, we are also interested to what extent such a gap can affect the performance on navigation. We re-evaluate the visual representations with the more VLN-fitting LXMERT (Tan and Bansal, 2019) language encoder

from Hong et al. (2021) apart from the results using RoBERTa. We show a full comparison between LXMERT and RoBERTa as the language encoder in Tab. 9.

Table 9: Exhaustive ablation on language encoder (RoBERTa *vs.* LXMERT) on navigation task. ResNet shows stronger fitting on seen scenarios, while ViT generalizes better on unseen scenarios.

| Encoder Pair (V+L) | Seen Succ. | Seen SPL | Unseen Succ. | Unseen SPL |
|---|---|---|---|---|
| RN-IN + RoBERTa | **54.75** | **51.02** | 48.62 | 42.61 |
| + LXMERT | 58.18 (+3.43) | 54.28 (+3.26) | 55.17 (+6.55) | 50.40 (+7.79) |
| RN-MoCo + RoBERTa | 52.50 | 48.92 | 48.49 | 43.49 |
| + LXMERT | 57.49 (+4.99) | 54.20 (+5.28) | 55.21 (+6.72) | 50.73 (+7.24) |
| RN-CLIP + RoBERTa | 52.40 | 48.39 | 49.00 | 43.49 |
| + LXMERT | 58.86 (+6.46) | 55.03 (+6.64) | 55.98 (+6.98) | 50.73 (+7.24) |
| RN-Ego + RoBERTa | 44.07 | 39.41 | 43.89 | 37.88 |
| + LXMERT | 45.45 (+1.38) | 41.89 (+2.48) | 48.45 (+4.56) | 44.00 (+6.12) |
| ViT-32-CLIP + RoBERTa | 51.03 | 46.98 | 49.21 | **43.99** |
| + LXMERT | 54.95 (+3.92) | 50.53 (+3.55) | 56.88 (+7.67) | 51.70 (+7.71) |
| ViT-16-CLIP + RoBERTa | 46.23 | 44.15 | 47.13 | 43.60 |
| + LXMERT | **61.90 (+15.67)** | **55.22 (+11.07)** | **59.51 (+12.38)** | **53.15 (+9.55)** |
| ViT-16-MAE + RoBERTa | 48.87 | 43.91 | **49.55** | 43.72 |
| + LXMERT | 48.29 (-0.58) | 43.16 (-0.75) | 52.41 (+2.86) | 45.99 (2.27) |

We observe that performances improve on nearly all visual representations when we use LXMERT instead of the text-only pre-trained RoBERTa. This demonstrates that visually-aligned language representations do massively improve downstream performances. And to further eliminate the influence of subsequent navigation decoder which is pre-trained along with the LXMERT language encoder, we conduct an extra ablation to test whether a from-scratch navigation decoder with randomly initialized weights, which loses the encoder-decoder pre-training alignment, can still achieve similar result. The results are shown in Tab. 10.

Table 10: Performance of LXMERT language encoder paired with pre-trained decoder *vs.* from-scratch decoder.

| Encoder Pair (V+L) | Decoder | Seen Succ. | Seen SPL | Unseen Succ. | Unseen SPL |
|---|---|---|---|---|---|
| ViT-CLIP-16 + LXMERT | pre-trained | 61.90 | 55.22 | 59.51 | 53.15 |
| ViT-CLIP-16 + LXMERT | from-scratch | 60.92 (-0.98) | 56.54 (+1.32) | 57.13 (-2.38) | 51.40 (-1.75) |

We observe that the model with a from-scratch navigation decoder still achieves comparable performance despite losing the encoder-decoder pre-training alignment. This shows that the improvements shown in Tab. 9 is not due to the encoder-decoder coupling, but rather due to the visually-aligned semantics of the LXMERT language encoder.

# E    SOCIAL IMPACT

With the emergence of foundation models, tremendous efforts have been put on improving multi-tasking capabilities of large-scale pre-trained models. Instead of improvements from the model's side, we focus on the problem of designing a general, sophisticated and fair evaluation metric that reveals the models' capabilities on the full spectrum of vision tasks. We believe our G-VUE benchmark will make an excellent complement to existing works as they can be easily adapted to be evaluated in G-VUE. There is no known negative societal impact at this point.

# F    LICENSE

We list the licenses of all dataset involved in G-VUE as follows:

- NYUv2: Unknown
- ShapeNet: Custom
- CambridgeLandmarks: CC BY-NC-SA 2.0 UK
- 7 Scenes: Custom
- Flickr30k: Custom
- RefCOCO: Apache License 2.0
- ADE20k: 3-Clause BSD License
- GQA: CC BY 4.0
- VCR: Custom
- Bongard-HOI: Custom
- R2R: Unknown
- Ravens: Apache License 2.0

