# OpenReview forum: "Perceive, Ground, Reason, and Act: A Benchmark for General-purpose Visual Representation"
_ICLR.cc/2023/Conference — Submitted to ICLR 2023_

### Official Review · Reviewer_U7v5 · 2022-10-14

**Confidence:** 3
**Correctness:** 2
**Technical Novelty And Significance:** 3
**Empirical Novelty And Significance:** 3
**Recommendation:** 5

**Clarity, Quality, Novelty And Reproducibility:**

Seems clear and original at least to this reviewer. A similar benchmark collection was proposed for videos (Liu et al 2020; https://openaccess.thecvf.com/content_CVPR_2020/papers/Liu_Violin_A_Large-Scale_Dataset_for_Video-and-Language_Inference_CVPR_2020_paper.pdf) -- but I'm not sure if something has caught on covering the proposed selection of tasks.

**Strength And Weaknesses:**

Strengths:
* Benchmarks are important, and to this reviewer the overall message of the paper (models ought to be evaluated on a variety of vision tasks, not just low-level vision for instance), seems reasonable.
* Making this leaderboard public will help other researchers compare to past work.
* The analysis of low-level encoders (ResNet50 versus ViT-B); with different pretraining mechanisms / data sources is interesting. I'm not quite sure how much it extends to much larger models, though, or models that use language through other means (besides just a source of "labels" like CLIP).

One overarching question to this reviewer is the purpose of this benchmark collection -- are researchers supposed to finetune models on it, evaluate zero-shot, or do something else (like linear probing)? In hindsight, looking at what happened in the text domain like GLUE, to this reviewer it seems like the zero-shot evaluation paradigm won out as many of the GLUE subtasks were quite noisy and easy to overfit to if the right finetuning recipe was applied. I'd be curious as to the thoughts of the authors here.

Weaknesses:
* The benchmark could be improved -- and the results in this paper more insightful -- if more models were chosen. I'd like to see how well OFA / UIO do on a finetuned setting, if that's what the benchmark is about (since the underlying image encoders are finetuned). If the message is that e.g. OFA can't do depth perception out-of-the-box (because it wasn't trained on that) -- I'd be curious as to how it would do in a linear-probe setting. I'd also like to see other vision-language models included to see the benefits of training on other types of data -- like perhaps FLAVA (Singh et al 2021 https://arxiv.org/abs/2112.04482 ) which is another visual-bert like model trained on slightly different data versus OFA, or MERLOT Reserve (Zellers et al 2022 https://arxiv.org/abs/2201.02639) trained on videos. Finally, I'd be curious as to whether the results are different with size (e.g. is UIO-XL better than a smaller Unified-IO model?)
* An issue with this collection of benchmarks, at least to this reviewer, is that they all have different formats (different inputs, outputs, metrics); different training distributions / validation sets, etc. For that reason, I'm not quite sure whether this benchmark collection will catch on. Though not as important for my overall score, I think it would be neat if some code was released that made it easy for different e.g. image encoders to be finetuned on the various tasks. However, I'm still not sure as to whether that code would be used -- as it would presume a model structure that might not be what's ideal for these various tasks.

**Summary Of The Paper:**

This paper presents a unified collection of 11 vision (and vision-and-language) benchmarks, called G-VUE. One of the messages of this paper is that there are a lot of different evaluations of vision models these days, and they often cover very disjoint things. The paper suggests that vision models be evaluated on a variety of different tasks, exposing different capacities. Some of the tasks are a bit simplified (in terms of their format) to make evaluation easier (e.g. VCR is just the question answering part of that benchmark).

The paper evaluates ResNets and Vision Transformers, pretrained on both imagenet, web images, and egocentric videos, and with objectives spanning supervised and self-supervised learning. These are often treated as underlying encoders in the literature, and here it seems that the self-supervised approaches do best -- CLIP is better for grounding/reasoning, and MAE is better for raw perception tasks like depth estimation. The paper also evaluates recent large-scale vision-language models proposed in the last year (OFA and UnifiedIO) and finds significant headroom on many of of the proposed tasks.

The G-VUE benchmark will have an associated leaderboard for other researchers to use.

**Summary Of The Review:**

To this reviewer, the introduction of a leaderboard that encourages evaluating on multiple tasks seems promising. However, it's not clear to this reviewer what the role of this leaderboard ought to be, and the analysis in the paper (i.e. a proxy of what we can learn from the leaderboard) could be improved by incorporating different classes of models.

---

> ### Author Response · Authors · 2022-11-10
> **Response to Reviewer U7v5**
>
> Thanks for your feedback. As you said, a holistic benchmark is important and reasonable for the evaluation of visual models. And such a public benchmark can benefit the community towards general-purpose vision. You also expressed your concerns, which we will address as below:
>
> 1. Q: *The purpose of this benchmark collection*
>
>     The core purpose is to evaluate visual representations in a fair setting. Generally, one visual representation cannot directly handle various downstream tasks, which require specific model structures. Hence, we equip the visual representation with several types of decoders and use the training data in each task to learn the corresponding decoder, while keeping the representation fixed, in order to ensure the performances on different tasks come from a common visual representation.
>
>     We show fine-tuning results in the appendix, but it is just for reference about how performance would change if we remove the fixed constraint on visual representations. Actually, we do not encourage task-specific fine-tuning, which goes against the spirit of general-purpose. On the other hand, zero-shot evaluation is impractical, because current visual representations are mostly learned from huge amounts of data and thus *unseen* becomes unclear.
>
>     To summarize, each collected task provides some data for the adaptation of visual representations. We also inject generalization test on some tasks, e.g., Navigation and Manipulation, where it is hard to benefit from pre-training. And we are aimed at evaluating visual representations instead of learning visual representations.
>
> 2. Q: *Extending experiments regarding more models*
>
>     The purpose of introducing the additional experiments on OFA/Unified-IO is not to explore their properties or make improvements. We just want to show that despite their remarkable versatility, they are still not general-purpose enough when exposed to a holistic evaluation of full-spectrum visual capabilities. Thus we advocate more dimensions of visual capabilities should be considered when learning visual representation. In addition, these models with cross-modal encoder do not exhibit neat visual representations, so systematic evaluation for  the visual representation is not so tractable.
>
> 3. Q: *The implementation of this benchmark*
>
>     Actually, the code is already released. A suite of tools including dataloaders and models can be found in the repo, and they are easy to use. The implementation of our framework introduces least inductive bias, so the comparison of visual representations can be fair and reasonable.

---

> > ### Comment · Reviewer_U7v5 · 2022-11-24
> > **thanks for the response! keeping my score**
> >
> > Thanks for the response! I'd like to keep my score, primarily because I think more models could be compared against here, and it would make for a much stronger contribution.

---

### Official Review · Reviewer_xbAo · 2022-10-17

**Confidence:** 3
**Correctness:** 2
**Technical Novelty And Significance:** 2
**Empirical Novelty And Significance:** 2
**Recommendation:** 3

**Clarity, Quality, Novelty And Reproducibility:**

Clarity: the writing is clear and detailed. Fig 3, the overview of the proposed model lies in the appendix, which is weird. The words in fig.1 are too small.

Quality: Some claims about the contribution, especially the model and benchmarks seem not suitable.

Novelty: weak. There are not many new insights and contributions of the general learning model, or benchmarks (metrics, data collection, annotation, etc.). Only some superficial conclusions and analyses are given.

Reproducibility: the authors have given the details of the proposed model and implementation, and all the datasets adopted are commonly-used, thus I think the reproducibility is OK.

**Strength And Weaknesses:**

Pros:
+ Pursuing the general purpose visual representation learning is important for our community.

+ Some results and discussions are interesting and verify some conclusions of previous works, e.g. the comparison between CNN and Transformer, and the superiority of CLIP.

Cons:
- There is no deep analysis of the relationship between the selected 11 tasks, only a performance correlation matrix is given. This is weird as the analysis of tasks is the most important thing to build a real unified benchmark, instead of simply compositing all existing sole datasets with domain gaps, different raw data, definition gaps, and annotation quality.

- Few new insights and discussions are provided, except for sec. 5.3. Moreover, "camera pose estimation and 3D reconstruction show fewer connections to other tasks", this seems weird, camera extrinsic parameter estimation matters most in 3D reconstruction in tasks like structure from motion. Please give a discussion.

- The claim about the proposed model is confusing. We these pre-trained encoders and fine-tuned decoders for these tasks in evaluation. But if we want to propose a new general model, what is the point of this model's existence? I think an experiment is suitable for this contribution instead of a model structure.

- Some discussion about Pathway from Google?

- ViT-B/32 MAE?

**Summary Of The Paper:**

This paper combines 11 tasks and benchmarks into a new benchmark for general-purpose visual representation learning. Four kinds of visual tasks are considered, i.e., perception, grounding, reasoning, and action. Then, by choosing different pre-trained backbones, a unified experiment is conducted to compare the learned representations from these backbones. Discussion and analysis are provided according to the results of the above investigation.

**Summary Of The Review:**

Overall, the experiments are extensive and implementations are detailed. However, besides the composition of previous datasets and tools cannot result in a "new benchmark", at least not enough. New insights into the benchmark design, the relation between previous datasets and tasks, and how to design a new general model are all lacking. I appreciate the efforts of this work. However, deeper discussions are essential. Moreover, the contribution claim about the model is also not suitable. I encourage the authors to make a major revision to make the analysis and discussion deeper and more inspiring. Thus, I think this paper is not ready yet currently for ICLR.

---

> ### Author Response · Authors · 2022-11-10
> **Response to Reviewer xbAo**
>
> Thanks for your feedback. We will answer your questions as below:
>
> 1. Q: *There is no deep analysis of the relationship between the selected 11 tasks. Few new insights and discussions are provided.*
>
>     I think our primary contribution is the perspective of evaluating visual representations, while deep analysis is not what we focus on. We conduct several baselines and report our findings. The relationship between tasks can be found in the benchmark design principles. If you mean quantitative relationships, this requires more observations.
>
> 2. Q: *"camera pose estimation and 3D reconstruction show fewer connections to other tasks"*
>
>     There might be a misunderstanding. The message we want to deliver is that these two tasks show fewer connections with other tasks, rather than a weak connection between these two tasks.
>
> 3. Q: *The claim about the proposed model is confusing*
>
>     We investigate a portion of current V-L models, and reach the balance between generality and effectiveness.
>
> 4. Q: *Some discussion about Pathway from Google?*
>
>     I think there is not much connection between our benchmark and Pathway. Pathway is more like a conceptual framework that leverages sophisticated engineering design with great capacity. However, it has not shown any practical application in general-purpose vision as far as I know.
>
> 5. Q: *ViT-B/32 MAE?*
>
>     There is no pre-trained MAE at patch size 32.

---

### Official Review · Reviewer_7KYb · 2022-10-24

**Confidence:** 4
**Correctness:** 3
**Technical Novelty And Significance:** 1
**Empirical Novelty And Significance:** 2
**Recommendation:** 3

**Clarity, Quality, Novelty And Reproducibility:**

Well-structured related work and clear paper presentation, however, the current section #3 reads as a set of separate tasks, while it would be great to focus on the alignment between visual cognitive abilities and the tasks for each functional domain, such that each task is better motivated. For example, why semantic segmentation is covered, but not scene-graph generation? Why visual QA and not embodied QA? There also seems to be limited (or none at all) tasks on compositional zero-shot learning and affordances, which are also fundamental human abilities. The Appendix covers *some* of these questions but it does not seem to be the right place for including the motivation for selecting specific tasks and datasets.

Figure 1 could be improved. It is unclear what the arrow between "Visual Representation" and the tasks represents.

Small grammar and typos here and there, e.g.,
"Architectures with too many complex ...", "Such findings further motivates the question ..."

**Strength And Weaknesses:**

Strengths:
- The benchmark will be made easily accessible through an evaluation platform and a leaderboard.
- The task correlation analysis and the decoders that can be used for evaluating any arbitrary model are nice pluses.

Weaknesses:
- It seems that the tasks are a composition of existing datasets with existing evaluation metrics. As such, the contribution seems limited.
- It is unclear whether the tasks in each functional domain are necessary and sufficient for evaluating general-purpose human-like vision systems, let alone if evaluating each functional domain separately is the best way to test human-like visual cognitive abilities.
- In Table 3, it is really difficult to extract from the summary score that ViT-16-CLIP is the best general-purpose model because, for some tasks, models have large deviations (such as I-T Retr.) while others do not. Would the underlying assumption of a summary average be sufficient, or should an overall evaluation metric consider task difficulty? Please also clarify the sentence "To make domains more balanced, we augment each task in Act with weight at 1.5".
- It is unclear how extensible the benchmark is, e.g., how easy it is in the future to include more tasks as newer datasets are constantly being constructed.
- Most evaluation does not consider multi-task models, but rather self-supervised or pretrained visual representations. It is a bit of a stretch to refer to this as a general-purpose vision benchmark in the conclusion and introduction (list of contributions). In addition, it would be great to have a direct comparison of a multitask model with the ones shown in Table 3. Adding this would demonstrate how far away self-supervised visual representation models are from finetuned multitask models.

A few additional questions:
1. In terms of the claim, that unified architectures are "still limited to the conventional vision-language domain, lacking the general capabilities for low-level geometric understanding and high-level reasoning and acting", what about GATO [1]?
2. Also, why are the unified architectures on Table 4 only evaluated on tasks they can solve, and not the full pool of benchmark tasks? Aren't the decoders described in this work specifically proposed for facilitating general evaluation purposes?
3. If the language encoder can be substituted with a CLIP language encoder, then what is the reason for not using this  CLIP language encoder for all models, and evaluating these models with improved cross-modal understanding capabilities?
4. How do the custom licenses for some of the datasets affect benchmark license and usage?

[1] Reed, Scott, Konrad Zolna, Emilio Parisotto, Sergio Gomez Colmenarejo, Alexander Novikov, Gabriel Barth-Maron, Mai Gimenez et. al. "A generalist agent." arXiv preprint arXiv:2205.06175 (2022).

**Summary Of The Paper:**

This paper presents a General-purpose Visual Understanding Evaluation (G-VUE ) benchmark that covers 11 tasks in four functional
domains (Perceive, Ground, Reason, and Act). The benchmark is accompanied by several decoders that allow the evaluation of arbitrary visual representation models on these tasks.

**Summary Of The Review:**

This paper aims to create a benchmark that can holistically evaluate the cognitive abilities of visual systems. This is work in the right direction, with functional domains properly articulated and motivated. However, the current implementation is a compilation of existing datasets, each with its own evaluation metrics. Improving the overall evaluation, for example by considering task difficulty, or a curriculum based on these tasks, or a unified evaluation metric and dataset, would make the realized work more aligned with the overarching goal of human-like general-purpose vision systems.

---

> ### Author Response · Authors · 2022-11-10
> **Response to Reviewer 7KYb**
>
> Thanks for your feedback. Your opinions are helpful to us. We answer them as follows:
>
> 1. Q: *It seems that the tasks are a composition of existing datasets with existing evaluation metrics. As such, the contribution seems limited.*
>
>     We clearly define four domains of visual capabilities and propose a comprehensive benchmark aimed at evaluation of general-purpose visual representation evaluation. The novelty lies in such a perspective, spanning from geometric perception, to embodied navigation and manipulation.
>
> 2. Q: *It is unclear whether the tasks in each functional domain are necessary and sufficient for evaluating general-purpose human-like vision systems, let alone if evaluating each functional domain separately is the best way to test human-like visual cognitive abilities.*
>
>     The rationality of our proposal can be found in the appendix.
>
> 3. Q: *Would the underlying assumption of a summary average be sufficient, or should an overall evaluation metric consider task difficulty?*
>
>     Though different tasks come at different levels of difficulty, how to identify and define such impact on the summary score remains unclear. And manual design regarding this issue is prone to bias. So we make this decision to avoid unnecessary bias.
>
> 4. Q: *It is unclear how extensible the benchmark is.*
>
>     Recruiting new datasets won’t affect the position of other tasks. You can check our repo if interested.
>
> 5. Q: *Most evaluation does not consider multi-task models, but rather self-supervised or pretrained visual representations.*
>
>     Yes, this is a piece of good advice. And we plan to evaluate some latest V-L pre-trained models with explicit visual representation.
>
> 6. Q: *About GATO*
>
>     GATO is an awesome work about general-purpose agent, which covers capabilities in vision, language, and robotics. Though, its main contribution lies in policy learning. Despite the capability to handle V-L inputs, it does not cover the skill of visual understanding tasks such as semantic segmentation.
>
> 7. Q: *why are the unified architectures on Table 4 only evaluated on tasks they can solve*
>
>     Because these unified architectures do not have explicit visual representation, which is required by the decoders to perform tasks.
>
> 8. Q: *what is the reason for not using this CLIP language encoder for all models*
>
>     Our initial assumption is that the evaluation of visual representation should be independent of the language branch. Particularly, a language encoder like RoBERTa has no relationship with the visual representations, which makes fair comparison, although CLIP text encoder leads to better performance due to the context of vision during pre-training.
>
> 9. Q: *How do the custom licenses for some of the datasets affect benchmark license and usage?*
>
>     There is no influence, since we do not make modifications or something else.
>
> 10. Q: *Tasks are not motivated well enough*
>
>     Due to the page limit, we put the motivations of selecting tasks and datasets in the appendix. In contrast to the motivations, the introduction and evaluation protocol of each task are more necessary for the main body, because they directly compose the benchmark (Section 3).

---

> > ### Comment · Reviewer_7KYb · 2022-11-24
> > **Post-rebuttal**
> >
> > Thank you for your responses. Due to my concerns about the contribution, the assumptions and scope details, and the benchmark design, I am inclined toward keeping my initial rating.
> >
> >     The rationality of our proposal can be found in the appendix.
> > Please provide further feedback on which pages/lines in the appendix. I am wondering if the appendix is a good placement for rationality and motivations of selecting tasks, especially if the rationality behind choosing specific tasks is dispersed across several parts of the appendix.
> >
> >     Recruiting new datasets won’t affect the position of other tasks. You can check our repo if interested.
> > The question on extensibility refers to how easy it is to add new benchmarks and datasets (it is clear that adding new datasets will not affect other tasks but nevertheless affect the summary average). As far as I understand, *there is no URL (e.g., GitHub page) that could be used to find authors’ identities*, so we cannot check the repo?

---

### Official Review · Reviewer_9fAp · 2022-10-26

**Confidence:** 4
**Correctness:** 3
**Technical Novelty And Significance:** 3
**Empirical Novelty And Significance:** 3
**Recommendation:** 5

**Clarity, Quality, Novelty And Reproducibility:**

The paper overall is clear and well-written. Some of the work in supplementary should move into the main paper.

**Strength And Weaknesses:**

[Strength]

- The proposed benchmark involves four main functions, which are perceive, ground, reason, and act. Seems like a good combination of skills to test the unified model.

- The author also proposes to use pre-trained representations to test the benchmark and evaluate two existing methods -- OFA and Unified-IO.

[Weakness]

- Creating a new benchmark is not simply aggregating existing datasets/skills. The more important for unified benchmark creation is what is the connection between each dataset, are there any shared concepts? what do we expect if the model performs well on one task and on others? Although the paper proposed interesting combinations of skills, there is no in-depth discussion on this.

-  The reason for choosing the dataset in each skill is not clear. For example, In question answering, the authors choose GQA dataset for this benchmark. It is known that GQA is created by question templates and a lack of language diversities. VQA might be a better option. For an evaluation benchmark, the best option would be to use a mixed of existing benchmarks for the testing set.

- The proposed framework uses frozen features and fine-tune the decoder on each task. The performance is far behind state-of-the-art results. Table 3 also doesn't have any STOA numbers on these tasks. My biggest concern is the trends shown in this table can not transfer or hint at other approaches.

- In table 7, even with the fine-tuning encoder, the performance is still behind the state-of-the-art model. By freezing the encoder and having a separate decoder for each skill, the proposed framework is not a unified model anymore.

- Table 4 shows two existing method results, OFA-HUGE and UIO-XL. These are interesting results, but many details are missing. For example, Is OFA-HUGE a single model or a model fine-tuned on each task? UIO-XL model didn't train on GQA datasets, this might be the main reason why the gap between OFA and UIO on VQA is huge. However, they may not reflect the general QA abilities since UIO also performed well on VQA datasets.

**Summary Of The Paper:**

This paper proposed a general purposed visual understanding benchmark (G-VUE), which covers four different abilities -- Perceive, Ground, Reason, and Act. The authors chose 11 existing tasks in this benchmark. The authors also provide a general encoder-decoder framework with pre-trained representations and tests on the benchmark.

**Summary Of The Review:**

The paper proposed an interesting benchmark to evaluate more unified models. However, there are multiple issues related to the connection between different tasks/datasets, and dataset selection for each skill. The problem of the proposed encoder-decoder model etc. Please check the weakness session for details.

---

> ### Author Response · Authors · 2022-11-10
> **Response to Reviewer 9fAp**
>
> Thanks for your feedback. Your opinions are helpful to us. We answer your questions as follows:
>
> 1. Q: *Discussion about connection between datasets*
>
>     Your suggestion is reasonable. However, due to the inhomogeneity of visual tasks, including various formulations, connection and in-depth discussion are not intuitive. Hence we do not put our concentration on this.
>
> 2. Q: *The reason for choosing datasets is not clear.*
>
>     We discuss our motivation of dataset selecting and the rationale in the appendix in detail. Specifically, comparing GQA and VQA datasets, we choose GQA since it has rich compositionality and is more challenging.
>
> 3. Q: *The performance is far behind SOTA... the trends shown in this table can not transfer or hint at other approaches.*
>
>     We admit the performances of baselines are far behind task-specific SOTA in some tasks. We implement the baselines with least inductive bias and task-specific design, which naturally leads to suboptimal performances. This is a trade-off between generality and effectiveness. Moreover, our baselines are a probe of the most typical visual representations, and the target is not to obtain high scores. Though, we select several common factors of visual pre-training, and the baselines are representative for these factors. So, despite being empirical, we think our findings could hint for others in terms of the discussed factors.
>
> 4. Q: *Limited improvements of finetuning… the proposed framework is not a unified model anymore.*
>
>     We conjecture the main reason for the gap with SOTA comes from large-scale pre-training as well as task-specific design. We do not claim we reach a unified model. Instead, we provide a unified framework for evaluating visual representations on our holistic benchmark composed of various visual tasks.
>
> 5. Q: *About Table 4*
>
>     OFA-Huge is a single model without finetuning here. In their paper, the OFA reaches SOTA after performing task-specific finetuning. The evaluation results of UIO on GQA dataset is really surprising, and our explanation is that the reasoning capability it acquires is associated with data distribution to some extent.

---

> > ### Comment · Reviewer_9fAp · 2022-11-23
> > **Thanks for the responses**
> >
> > The author's response partially solves my concerns about this paper. However, I still think there are issues in terms of the benchmark and baselines.

---

### Decision · Program_Chairs · 2023-01-20

**Decision:**

Reject

**Justification For Why Not Higher Score:**

All reviewers felt this paper did not meet the bar for ICLR. Although reviewers commended the effort to create a benchmark for general-purpose vision, they did not feel the current framework of 11 existing tasks contributes substantially enough beyond what already exists. Reviewers also felt that tasks felt a bit disconnected and did not get clear insights beyond what we already know from the benchmarks for each task on its own. The authors engaged in discussion, and the reviewers considered the rebuttals, but were not swayed. The AC agrees with this consensus and advocates rejection.

**Justification For Why Not Lower Score:**

N/A

**Metareview: Summary, Strengths And Weaknesses:**

Summary:
This paper proposes a meta-benchmark for evaluating visual representations. The meta-benchmark consists of 11 tasks from previous benchmarks. To evaluate a visual representation (i.e. an image encoder), the representation is first decoded into an appropriate format for each of the 11 tasks, then evaluated on that task. Findings include that transformers and language supervision outperform alternatives.

Strengths:
* Such a meta-benchmark could be useful as we move toward ever more general vision systems.
* Evaluating with task-specific decoders is a nice way to support testing a variety of encoder models.

Weaknesses:
* The contribution is limited to a meta-framework on top of 11 existing benchmarks.
* Could benefit from making stronger connections across the tasks, or a single metric to summarize overall ability.
* The benchmark focuses mainly on image encoders (representation learners). Aside from Section 5.4, it does not evaluate multitask models that directly perform some or all of the 11 tasks.
* Only a few methods are compared.